# Spanish Facebook Posts as an Indicator of COVID-19 Vaccine Hesitancy in Texas

**DOI:** 10.3390/vaccines10101713

**Published:** 2022-10-14

**Authors:** Ana Aleksandric, Henry Isaac Anderson, Sarah Melcher, Shirin Nilizadeh, Gabriela Mustata Wilson

**Affiliations:** 1Multi-Interprofessional Center for Health Informatics, The University of Texas at Arlington, Arlington, TX 76019, USA; 2Department of Computer Science and Engineering, The University of Texas at Arlington, Arlington, TX 76019, USA; 3The University of Texas at Arlington, University Analytics, Arlington, TX 76019, USA

**Keywords:** social media, sentiment, vaccine hesitancy, public health, interventions

## Abstract

Vaccination represents a major public health intervention intended to protect against COVID-19 infections and hospitalizations. However, vaccine hesitancy due to misinformation/disinformation, especially among ethnic minority groups, negatively impacts the effectiveness of such an intervention. The aim of this study is to provide an understanding of how information gleaned from social media can be used to improve attitudes toward vaccination and decrease vaccine hesitancy. This work focused on Spanish-language posts, and will highlight the relationship between vaccination rates across different Texas counties and the sentiment and emotional content of Facebook data, the most popular platform among the Hispanic population. The analysis of this valuable dataset indicates that vaccination rates among this minority group are negatively correlated with negative sentiment and fear, meaning that a higher prevalence of negative and fearful posts indicates lower vaccination rates in these counties. This first study investigating vaccine hesitancy in the Hispanic population suggests that observation of social media can be a valuable tool for measuring attitudes toward public health interventions.

## 1. Introduction

On 11 March 2020, the World Health Organization (WHO) officially declared COVID-19 a pandemic [1]. Almost a year later, the United States Food and Drug Administration (FDA) granted the first emergency use authorizations for COVID-19 vaccines [2]. Since then, COVID-19 vaccines have become more widely available to the public, and data on case counts, deaths, and vaccinations have been tracked across the country. However, public acceptance of the vaccine is crucial in effectively combating the pandemic [3]. Regardless of vaccine availability, vaccination rates have been impacted by access to accurate health information and the political polarization of the pandemic.

Multiple studies and public opinion polls have assessed vaccine hesitancy throughout the pandemic. Khubchandani et al. [4] surveyed adults from numerous communities across the US and found that while most respondents expressed a willingness to get a COVID-19 vaccine, non-White and lower socioeconomic respondents were less willing to be vaccinated. The Kaiser Family Foundation (KFF)’s Vaccine Monitor Project polled individuals about their vaccine willingness between December 2020 and February 2022 [5]. The number of respondents who said they would “wait and see” before getting a vaccine steadily declined during this period, but the number of respondents who said they would “definitely not” get the vaccine stayed consistent. Furthermore, multiple studies suggest that confidence in COVID-19 vaccines is low, revealing a distrust toward vaccine safety, effectiveness, and the healthcare system [6,7,8,9,10,11,12].

Multiple studies have also analyzed the sentiments of Twitter posts (tweets) in the context of the pandemic [13,14,15,16,17]. Sentiment, in this context, is a measure of an author’s feelings toward a given topic. Sentiment is often measured using sentiment polarity, where a document is rated as being “positive”, “negative”, or somewhere in between (e.g., a numeric value between 0 and 1, where 0 is negative and 1 is positive). For example, the average sentiment of COVID-19 vaccine-related tweets strongly correlates with vaccination rates at the state level [18]. This study also establishes a connection between social determinants of health (SDoH) and vaccination rates in the United States [18]. Monselise et al. [19] studied public sentiment regarding COVID-19 vaccinations within the first 60 days of their availability. Their findings suggest that the major concerns regarding COVID-19 vaccines are vaccine administration and access, while the most prevalent emotion found in the tweets was fear. In addition, many studies examined emotions contributing to vaccine hesitancy. The results indicate that the major reasons for individuals not getting vaccinated are concerns about the adverse effects of the vaccines and fear of vaccine safety [20,21,22]. Therefore, detecting emotions such as fear in social media posts regarding COVID-19 vaccines could provide insights into the public acceptance of the vaccine, and help to identify populations expressing more anxiety towards the vaccine who need further interventions to reduce vaccine hesitancy.

The current paper investigates similar questions to the above work, but with several important differences. First, the focus of this study is county-level data within Texas, rather than state-level or country-level data, providing granular insights into the specific locations. Moreover, 39.6% of Texas’s total population is Hispanic, and this population has shown the highest COVID-19 infection and death rates [23]. This disproportionate impact indicates potential shortcomings in existing public health infrastructure, which the current work aims to address. The second major novel component of this study is the analysis of Facebook [24] data, the most popular platform used by the Hispanic population [25], which is annotated for sentiment (positive or negative) and emotional content (fear, joy, and anger) using modern natural language processing (NLP) tools. The previously mentioned studies [18,19,20,21,22] have shown that sentiment strongly correlates with vaccination rates, but there is little work studying emotional content. The current study examines whether emotional content, such as joy and anger, shows any correlation with vaccination rate, as this greater level of detail may prove more useful for informing public health interventions. The third major contribution of this work is that the data were exclusively collected and analyzed in Spanish, while nearly all existing research focuses on collecting English text or relies on translating Spanish text into English before performing any analyses. Therefore, this is the first study investigating vaccine hesitancy in the Hispanic population by analyzing social media data in Spanish.

The current analysis is focused on testing the following hypotheses:

**H1:** 
*Negative social media sentiment and vaccination rates are negatively correlated at the county level in Texas.*


**H2:** 
*Positive social media sentiment and vaccination rates are positively correlated at the county level in Texas.*


**H3:** 
*Fear expressed in social media posts is negatively correlated with vaccination rates at the county level in Texas.*


**H4:** 
*Joy expressed in social media posts is positively correlated with vaccination rates at the county level in Texas.*


**H5:** 
*Anger expressed in social media posts is negatively correlated with vaccination rates at the county level in Texas.*


## 2. Materials and Methods

### 2.1. Data Collection

This study focuses on data from a widely used social media platform, Facebook [24], as the largest percentage (72%) of the Hispanic population is using Facebook compared to other social media platforms [25]. The data consist of 25,314 Spanish vaccine-related posts posted within Texas between 1 January and 31 December 2021. At the end of this time period (end of 2021), a large portion of the population had already been vaccinated. Thus, the vaccine hesitancy peak was likely soon after the vaccine administration started in 2021. Therefore, this study explores the sentiments and emotions of social media posts for the first 12 months after vaccination began. Posts were collected using the CrowdTangle tool [26], using a set of vaccine-related search terms. Figure 1 summarizes the Facebook data collection pipeline. The initial step of data collection is finding a set of keywords to search for vaccine-related posts. The second step involved using CrowdTangle to query posts containing any of the keywords from the established list of keywords. Afterward, emotions and sentiments were obtained for each post in the dataset. The final step is the statistical analysis used to test the formulated hypotheses. These steps are discussed in more detail in the sections below.

County-level vaccination data were obtained from Texas COVID-19 Hospital Resource Usage and Vaccinations [27], which collects and reports daily vaccinations from the Texas Department of State Health Services website [28]. County-level demographic data (percentages of Hispanic, Black, and Asian population per county) were obtained from the American Community Survey (ACS) demographic and housing estimates of the US Census Bureau [29].

#### 2.1.1. Facebook Data Collection

The first step of Facebook data collection was to create a set of keywords to use as search terms in CrowdTangle, in order to identify vaccine-related posts. Filtering by these keywords before performing the text analysis allowed the analysis to target authors’ sentiments, specifically towards vaccines and vaccine-related topics. Keywords were selected using a snowball technique, conducted in consultation with a native Spanish speaker, where a set of “seed words” were selected which the speaker identified as highly associated with vaccine-related discourse. Then, Facebook posts containing these words were monitored, and any new words or phrases that appeared with high frequency were added to the set of keywords. This process was repeated until a large set of keywords was identified, and all relevant keywords appearing in the posts were included in the list. The final keyword list is available in Appendix A.

The second step of data collection was using CrowdTangle to query posts containing any of the identified keywords. CrowdTangle allows for selecting posts based on keywords, language, local relevance, and time period, as well as the social media platform, specific pages or groups, etc. The data provided by CrowdTangle include messages posted on influential public pages with more than 25,000 page likes or followers. However, CrowdTangle does not provide posts that are shared by regular Facebook accounts. Even though the data contain certain limitations, such data still provide useful insights into public attitudes about certain topics which people discuss on public Facebook pages. Therefore, the keyword-related posts were obtained for the entire year of 2021 by selecting the top 20 Texas counties with the highest number of COVID-19 cases in December 2021. In more detail, a single search was performed by choosing Spanish posts based on the created set of keywords shared within the pages associated with one of the counties. Then, the obtained posts were marked as belonging to that one specific county. This process was repeated 20 times (corresponding to the number of counties selected for the analysis). Using this approach, the analysis clarified which posts were associated with which geographical region.

The third step involved finding emotions and sentiments for each post in the dataset. This was done using the pysentimiento Python library [30], which provides Spanish-language models for sentiment analysis (positive, negative, and neutral) and emotion detection (joy, sadness, anger, surprise, disgust, and fear). Pysentimiento provides a set of BETO [31] models that were fine-tuned for the respective classification tasks. Linear regression models with clustering of standard errors were used to investigate the relationships between these annotations and the county-level vaccination rates.

#### 2.1.2. Dataset Size

Initially, twenty counties with the highest number of COVID-19 cases in December 2021 were used to collect Facebook data. However, eight counties yielded a very low number of posts, and they were excluded from the analysis. Only counties with at least 100 posts were included, leaving a total of 12 counties and 25,314 posts. The study includes Harris, Dallas, Tarrant, Bexar, Travis, El Paso, Fort Bend, Hidalgo, Lubbock, Webb, Cameron, and Nueces (counties are listed in order of the highest number of COVID-19 cases to the lowest number of cases). Figure 2 highlights the counties included in the study (circled in green), where the size of blue circles corresponds to the number of COVID-19 cases per county in 2021 (dashboard developed by the Texas Department of State Health Services [32]). Examples of posts in our dataset can be found in Appendix B.

CrowdTangle reports a location for each post based on local relevance rather than the location of the user or page where the post was made. Local relevance is based on the locations of users who follow a page. Thus, for a page with many followers in Houston, Texas, posts made to this page will show up as locally relevant to Harris County, where Houston is located. If the page has many followers from different Texas regions, e.g., Houston, Dallas, and El Paso, the same post will be listed once for each county. This causes a large number of posts to be seemingly duplicated, i.e., the same post is listed across several counties. Additionally, the dataset contains many instances of “re-posts”, where the exact same text is re-posted by different accounts, sometimes in the same county, sometimes in different ones. Based on a manual inspection of the data, many of these posts appeared to be news and media agencies re-posting the same message across different local affiliate pages. The most extreme example of such duplication was a post about the experiences of cancer patients with COVID-19 vaccines, which was posted 88 times across several different NoticiasYa pages (NoticiasYa is a Spanish-language local news portal), e.g., NoticiasYa Lubbock (in Lubbock, Texas), NoticiasYa 48 (in Corpus Christi, Texas), and the main NoticiasYa page.

Approximately 28.4% of all posts were duplicates from other counties, and about 32.3% were repeated within the same county. No duplicate posts were removed from the dataset prior to the analysis. Since the analysis targets individual counties, the same post appearing across different counties does not pose an issue. The same post may be viewed by users in multiple counties and lead to potentially different county-level effects. Duplicates of a single post appearing within the same county were also kept in the dataset, since the large user population on Facebook makes it unlikely that a duplicate post will necessarily result in duplicate exposure to different users, and the majority of such duplicates only appeared twice. This sort of re-posting is also common behavior on social media; thus, it represents an authentic component of social media discourse. The current work is also focused on developing social media monitoring techniques. The ultimate goal is to identify practical and useful predictors of county-level vaccine hesitancy; removing duplicate posts is not considered strictly necessary unless their inclusion has demonstrable negative impacts on the ability to identify useful signals. A histogram of the number of times post content was repeated in the dataset can be found in Figure 3.

### 2.2. Dependent, Independent, and Control Variables

This study focuses on finding the correlation between the vaccination rate at the county level in Texas and the sentiments and emotions of Facebook posts posted within these counties. Therefore, the dependent variable in this study is the weekly vaccination increase, while the independent variables are the sentiment or emotions of Facebook posts. However, other factors, such as social determinants of health, county demographics, and health literacy, are likely to be strongly predictive (or be strong proxies for other predictors). Therefore, health literacy, social vulnerability index, and county race and ethnicity composition were included as control variables in statistical models, in order to better isolate the contributions from sentiment and emotional content in the Facebook posts.

**The vaccination increase (rate)** is used as the dependent variable in the regression analysis, and is calculated as the weekly increase in the total number of doses distributed per county (e.g., the vaccination increase of week 2 is calculated by subtracting the total number of vaccinations in week 1 from the total number of vaccinations in week 2). Since the cumulative vaccination rates are monotonically increasing (an individual cannot be “un-vaccinated”), using the cumulative rates as the dependent variable risks measuring the wrong outcomes, e.g., it risks measuring how many people have ever been vaccinated as a function of current social media trends, which is both nonsensical and not especially useful for public health officials. Figure 4 shows the weekly vaccination increase per county, displaying a similar trend in almost all counties in the dataset, with the peak in vaccination increase in week 15 (corresponding to dates between 12 April and 18 April 2021). The higher peaks might reflect that on 15 April 2021, the Pfizer-BioNTech vaccine was authorized for ages 16 and up, while the Moderna and Johnson and Johnson vaccines were authorized for ages 18 and up [33].

**Sentiments and emotions** of all posts were obtained by using pre-trained transformer neural network models, trained for sentiment analysis and emotion detection in Spanish. Transformer models currently provide some of the highest accuracies for a wide range of language tasks, such as sentiment and emotion detection, and have already been applied with promising results to social media posts about COVID-19 [34,35,36]. This work uses the trained models provided by the pysentimiento Python library [30]. Pysentimiento’s Spanish models are fine-tuned instances of BETO [31]. The library’s authors fine-tuned BETO on the TASS dataset [37] for three-way sentiment classification (positive, negative, and neutral), and reported a micro-averaged F1 score of 0.672 on a held-out test set. They also fine-tuned a separate copy of BETO on the EmoEvent dataset [38] for six-way emotion classification (anger, disgust, fear, joy, sadness, and surprise), and reported a micro-averaged F1 score of 0.688 on a held-out test set.

The pysentimiento Python library contains a function that performs the cleaning of Spanish text by replacing user handles (@usernames) and URLs, shortening repeating characters, normalizing laughter, and handling hashtags and emojis. This function has been used to clean the posts in the dataset before passing them to sentiment and emotion detection models. Then, both models provide continuous numeric values between 0 and 1 for each category, such that all values sum to 1. For example, the sentiment model may assign a post a score of 0.2 for negative sentiment, 0.7 for positive sentiment, and 0.1 for neutral sentiment. In light of this, the neutral sentiment category is omitted from the current analysis. The emotions of disgust, sadness, and surprise were also omitted, since there was extremely little variance in these scores, with the vast majority of posts assigned nearly 0.

Table 1 contains basic descriptive statistics for the models’ predictions on the collected Facebook posts. Categories omitted from the later analysis steps are included for completeness.

While the performance of these models is not necessarily state-of-the-art, there are still several reasons to use them in this work. First, there are very few off-the-shelf sentiment and emotion classification models available in Spanish, and most of the existing ones are either trained exclusively on Castilian Spanish; the variants of Spanish used in Texas are divergent enough from Castilian that there are potential concerns about the validity of using such models in this setting. Second, the models are often trained on linguistic domains such as movie reviews, which represent a very different set of registers from most social media posts. By contrast, the TASS and EmoEvents datasets are derived from social media sources (primarily Twitter), and contain a wider range of Spanish variants. Second, while the reported F1 scores may appear low, they are still notably better than a random guess (which would give an F1 score of 0.5). Since the current work focuses on a fairly large corpus and population-level trends, the reported performance should be adequate to identify major trends. As such, the models’ performance metrics are not seen as a critical issue for this work, although they do represent an area for improvement and future work.

In order to visually represent the relationship between sentiments and vaccination increase, the average weekly vaccination increase was calculated, including all the weekly vaccination rates for all counties in the dataset. Similarly, average weekly negative and positive sentiments were computed. The average weekly increase in vaccinations is plotted with respect to the average weekly positive and negative sentiments in Figure 5. Larger and darker bubbles correspond to the higher average weekly increase in vaccinations, while lighter and smaller bubbles represent a lower average increase in weekly vaccinations. A higher average negative sentiment per week corresponds to lighter/smaller bubbles. In comparison, a lower weekly average negative sentiment corresponds to darker/larger bubbles (greater weekly average change in vaccination rate).

Weeks with the highest average negative sentiment presented in Figure 5 are weeks 31 and 29, corresponding to the time between 19 July and 8 August 2021. During this time, there was a significant increase in COVID-19 cases due to the Delta variant and breakthrough infections in Barnstable County [39]. In addition, on July 21, new research was released showing that Pfizer and AstraZeneca vaccines were less effective against the Delta variant than other variants [40]. Furthermore, at the end of July, it was shown that individuals who were infected with COVID-19 in the past were more likely to experience adverse events following immunization with the Pfizer vaccine [40]. These events likely contributed to additional stress in certain communities, triggering higher average negative sentiment during this time interval. This provides good evidence for the argument that observation of social media can give detailed insights into communities’ attitudes in certain areas related to ongoing events.

**Health literacy (HL)** refers to the ability of individuals to find, understand, and use the information necessary to perform required health-related decisions [41]. A recent study [18] showed a statistically significant positive correlation between HL and vaccination rate at the state level in the US, indicating that the vaccination rate is higher in states where HL is higher. Thus, the ability to access and understand health-related information may have a strong impact on vaccination rates at the county level as well—i.e., individuals with very low health literacy may rely more on misinformation about the vaccines or simply not know how to access reliable information sources. Therefore, HL is included as a control variable in statistical regression models. HL data were obtained from the University of North Carolina at Chapel Hill [42], based on 2010 census block data. The weighted average of HL, using the population of census block groups as weights (also from 2010), was calculated to obtain HL data at the county level.

**Social vulnerability index (SVI)** indicates the extent to which a community experiences negative effects when external stresses (such as a pandemic) are applied [43]. Areas with a higher SVI indicate more severe negative outcomes experienced in such areas. Furthermore, a related study suggests a significant relationship between SVI and the vaccination rate at the state level in the US, indicating that the vaccination rate is lower in more vulnerable areas [18]. In addition, researchers found an efficient way to identify high-risk areas by using the Health Intelligence Atlas [44]. This surveillance tool can help public health authorities plan required interventions by monitoring vulnerable communities and their COVID-19 vaccine hesitancy on the county or state level in the US [44]. The SVI scores for the twelve counties included in the datasets were extracted from the Centers for Disease Control and Prevention (CDC/ATSDR SVI) database that helps emergency response planners and public health officials identify at-risk communities [43]. These values were used as a control variable in the present analysis, as SVI might play a significant role in the Texas vaccination rate at the county level.

**Race and Ethnicity Composition** consists of the percentages of Hispanic, Asian, and Black populations in each county. These factors are likely to be strong proxies for many factors related to HL and SVI, and are included as additional control variables.

A summary of all of the data sources is presented in Table 2.

### 2.3. Statistical Analysis

The final dataset consists of Spanish Facebook posts, where each post is associated with the county where it has been posted, county-level HL, county-level SVI, and county-level race and ethnicity composition, as well as the week of the year when the post was published, and the vaccination increase for that week in that county. Furthermore, the dependent variable (vaccination increase) represents a count variable, leading to the usage of the linear regression model in the analysis. Standard errors were clustered by county and week due to a large amount of repetition in county-level and week-level features, such as HL, SVI, race and ethnicity composition, and vaccination rate. Five different regression models were used to evaluate five hypotheses. These were performed by changing the independent variables, namely negative sentiment, positive sentiment, fear, joy, and anger, for H1–H5, respectively. All control variables were included in the models.

## 3. Results

### 3.1. Sentiment Analysis Results (H1–H2)

A linear regression model was employed in order to evaluate the first hypothesis, including clustering standard errors by county and week. The dependent variable was vaccination increase; the independent variable was a negative sentiment; and HL, SVI, and demographic features were used as control variables. Model results show a strong, negative, statistically significant correlation between vaccination increase and negative sentiments (*p* < 0.05). These findings support Hypothesis 1, indicating that the larger negativity of the posts shared within a county is correlated with the lower weekly vaccination increase. The control variables included in the model were not statistically significant. The results of the model can be found in Table 3.

Testing H2 required a similar model, using positive sentiments instead of negative sentiments as the explanatory variable. The results did not show a significant correlation between positive sentiment and vaccination increase. The control variables remained unsubstantial in the model.

### 3.2. Emotion Analysis Results (H3–H5)

Testing H3–H5 included a linear regression analysis by clustering standard errors per county and week in order to find the association between the dependent variable, vaccination rate, and each independent variable (fear, joy, and anger scores). A separate model was created for each independent variable in order to test each respective hypothesis. All control variables were included. The results suggest a statistically significant negative correlation (*p*-value < 0.001) between fear and vaccination increase at the county level in Texas, meaning that a higher level of fear expressed in Spanish Facebook posts is associated with a lower vaccination rate in Texas, confirming H3. The results of the model testing H3 are displayed in Table 4.

Testing H4 and H5 did not yield a statistically significant correlation between joy and vaccination rate or between anger and vaccination rate (*p*-value > 0.05).

In order to further examine the anger results, 100 random samples of the posts wherein the predominant emotion was anger (i.e., where the post’s highest emotion score was for anger) were extracted. Then, two Spanish speakers and one English speaker (who translated the texts into English) annotated each post to identify the object of the poster’s anger. The annotators identified topics such as the vaccine itself, the government, individuals “cheating the system”, unvaccinated individuals, vaccine availability, and others (which do not fit in any of the previous ones) as the most common. The results suggest that in 42% of the posts, anger was directed at using the vaccine to “cheat the system”. These posts discuss events such as getting vaccinated without following the immunization schedule established by the CDC (e.g., getting vaccinated before the elderly population got vaccinated).

Furthermore, in 19% of the posts, anger was directed towards the government entities. In only 8% of posts was anger directed at the vaccine itself. This may provide some explanation for the unexpected results: individuals who express anger when discussing vaccines do not appear, in general, to be directing that anger towards the vaccine itself, or vaccination efforts more generally. These expressions of anger are, thus, less likely to indicate vaccine hesitancy or distrust.

A similar methodology was used to label joyful posts manually. In 42% of these posts, joy was directed toward vaccine availability, e.g., by mentioning where they are being distributed free of charge at various vaccination sites. In 22% of the posts, joy was toward the vaccine itself, while in 32% of joyful posts, joy was directed toward other topics that did not share any obvious common connection. Thus, such numbers provide an explanation for the lack of a significant correlation between joy and vaccination rate in Texas. Only approximately one-fifth of the posts were joyful about vaccination itself.

## 4. Discussion

This study investigated the association between sentiments and emotions of social media posts and COVID-19 vaccination rates in Texas, while controlling for other social determinants of health, such as HL, SVI, and county demographics. The results support two of the five initial hypotheses of this study, as Spanish Facebook data demonstrated that negative sentiment and fear both have statistically significant, negative correlations with vaccination rate increases. These results make a strong argument for public health officials to make more extensive use of social media data to monitor vaccine hesitancy, and to conduct culturally sensitive interventions tailored to the needs of each community. It also reinforces the recommendations from the WHO, which state that social media should be used to disseminate correct health information and counter misinformation [45]. The results presented in this study show that in counties where vaccination rates increase more slowly, social media posts express a more negative sentiment when discussing vaccine-related topics. Such results may imply that posts with negative sentiments strongly impact individuals’ feelings about the vaccine. In addition to social determinants of health and health literacy, particular psychological determinants, such as beliefs and attitudes, should be integrated into future studies, as the relationship between misinformation and preventive behavior adherence also plays a role in COVID-19 vaccine hesitancy [46,47,48]. In addition, counties with a lower vaccination rate showed a higher amount of fear directed towards the vaccine, which may result from misinformation and conspiracy theories which are easily shared over social networks. Thus, health officials can identify the sources of fear among certain communities and increase their online and offline intervention efforts. Therefore, observation of social media should become an active strategy to combat vaccine hesitancy and increase trust in the public health system. Furthermore, social media can be utilized to combat barriers of negative sentiment and fear by sharing encouraging posts related to the vaccine, as well as accurate and reliable information sources, in order to increase health literacy in certain geographic regions. Target messaging could potentially become a successful strategy to reach vaccine-hesitant populations on social media and attempt to change their points of view in terms of their vaccination decisions.

The results did not show a significant association between positive sentiment, joy, or anger and vaccination increase, indicating that negative sentiment and fear are the most impactful independent variables in the presented models. Qualitative analysis of the posts revealed that anger is not often directed towards vaccines and vaccination efforts, but rather at the government, people manipulating the system, etc., which might be a reason why it did not show a significant association with the vaccination rate in Texas. Similarly, joy might be directed not only toward vaccines but towards vaccine availability and other topics.

The previous study [18] that analyzed Twitter activity in English showed that the social vulnerability index and health literacy are also associated with COVID-19 vaccination rates. In this study, which analyzes Facebook social media activity in Spanish, HL is positively associated with COVID-19 vaccination rates, but the relationship is not statistically significant. This might be explained by the fact that HL data were calculated in 2015 based on 2010 census data [42]. The Hispanic population in Texas has increased rapidly since then [49]; however, HL has not been updated to reflect these changes. Similarly, SVI is negatively associated with the vaccination rate, but this is not statistically significant either. This may be due to the low number of counties used in the analysis and the high degree of repetition in SVI and HL measurement; the lack of variation compared to the post-level features may contribute to the lack of statistical significance.

The main strength and novelty of this study are the utilization of social media and the focus on Spanish language posts, which provide useful insights into individuals’ opinions about ongoing public health interventions (i.e., vaccination). However, one of the limitations of this study is that datasets contain only publicly available posts shared within public pages, and the analysis was limited to posts with explicit geolocation information. Natural extensions of this work would include more extensive data collection and additional geolocation strategies, e.g., incorporating user-level rather than just post-level geolocation information. In addition, obtaining large amounts of social media data at more granular levels, such as census block or zip code level, is rarely feasible. However, such data would permit a more detailed analysis of the relationships between SDoH and social media sentiments/emotions. Future work may include more detailed investigations of online discussions surrounding vaccinations, such as the spread of misinformation and disinformation, and more detailed analyses of the sentiment and emotion scores.

An important extension to this work would involve addressing the accuracy of the sentiment and emotion detection models. While the models from pysentimieno are sufficient for the current analyses, their accuracy leaves much room for improvement. Hand-coding a subset of the collected Facebook posts and training or fine-tuning a custom model would likely offer additional accuracy and robustness, allowing more accurate monitoring of online discourse surrounding public health issues.

## 5. Conclusions

This first study investigating vaccine hesitancy in the Hispanic population demonstrated an association between negative sentiments of social media posts in Spanish and COVID-19 vaccination rates, as well as the association between emotions such as fear and vaccination rates at the county level in Texas. Therefore, active observation of social media could become an effective tool for identifying communities which express higher levels of vaccine hesitancy and anxiety toward the vaccine. This information can be used to develop and target health interventions for communities that express negative sentiments or high fear towards vaccination. Such public health interventions could potentially increase vaccination rates in these areas and reduce vaccine hesitancy by using specific messaging which could raise awareness about the benefits of vaccination. Furthermore, this approach shows that, aside from analysis of English data, in certain geographical regions where a large percentage of the population is Hispanic, an effective way to reach such people would be to analyze the data and share targeted messages in their native language. Such an approach could be more successful in educating this population and reducing vaccine hesitancy in certain regions. While this study contains certain limitations, it provides opportunities for future work in this field. For example, the data provided in the study were collected from influential public pages, which do not provide insights into the social media activity of private or personal accounts.

Moreover, the sentiments and emotion detection models of Spanish text could be further improved. Despite the limitations of this study, the models used were still able to sufficiently detect sentiments and emotions that were used to find correlations between vaccination increase and negative sentiment, as well as fear, in Texas. Vaccine hesitancy will remain a major problem that needs to be addressed at the local, national, and international levels. Further analysis of social media and similar data sources will be crucial to combating vaccine hesitancy in the long term. Such analysis is essential for public health professionals and community partners to develop timely, culturally sensitive, and strategic messages to address the vaccine information needs of each community they serve.

## Figures and Tables

**Figure 1 vaccines-10-01713-f001:**
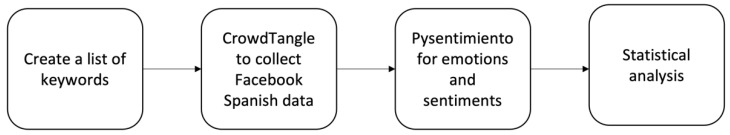
Facebook data collection workflow.

**Figure 2 vaccines-10-01713-f002:**
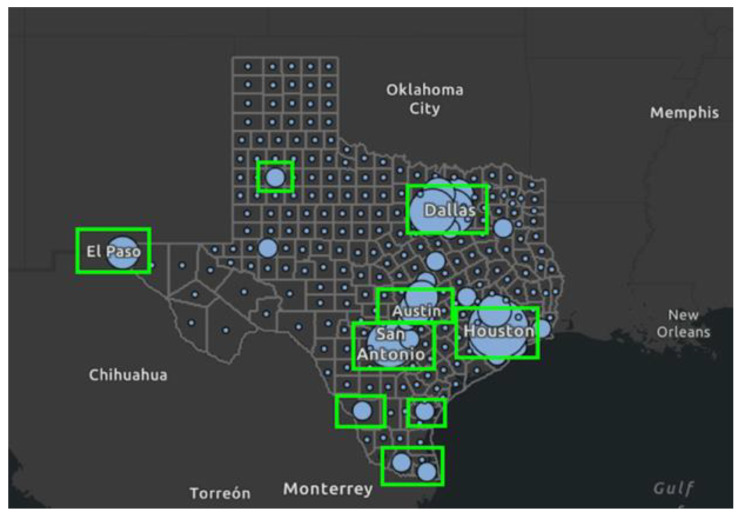
Texas counties included in the study are highlighted in green (dashboard developed by the Texas Department of State Health Services [32]).

**Figure 3 vaccines-10-01713-f003:**
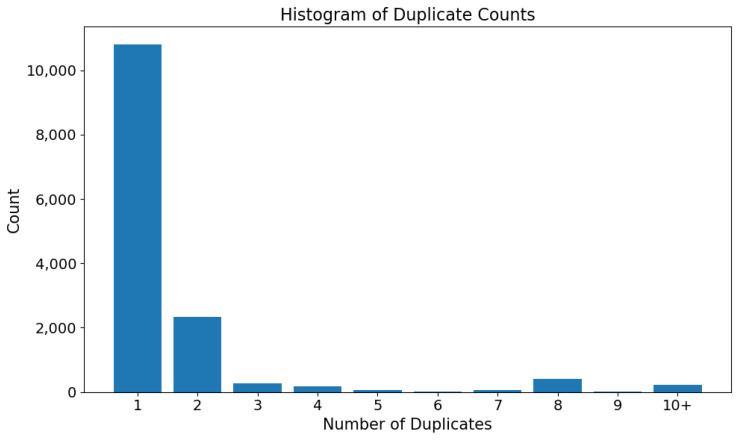
Histogram of the total number of repetitions of the post content in the dataset. Duplicates are counted as posts with identical texts.

**Figure 4 vaccines-10-01713-f004:**
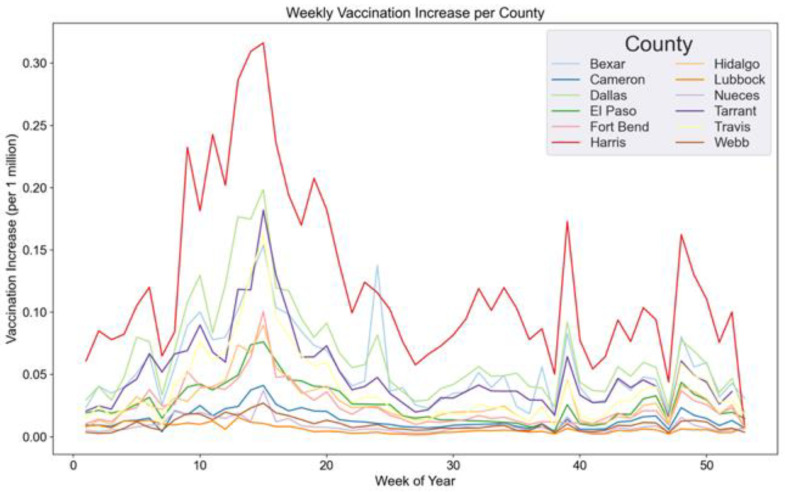
Weekly vaccination increase per county.

**Figure 5 vaccines-10-01713-f005:**
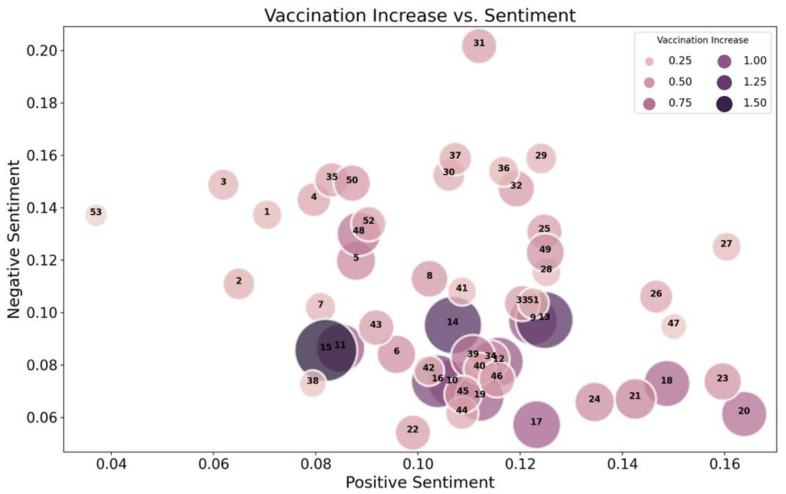
An average weekly vaccination increase with respect to average weekly positive and negative sentiment. Larger and darker bubbles correspond to a larger vaccination increase per week.

**Table 1 vaccines-10-01713-t001:** Descriptive statistics of sentiments and emotions.

Variable	Min	1st Quartile	Median	Mean	3rd Quartile	Max
Positive Sentiment	0.0003	0.01	0.02	0.08	0.06	0.99
Negative Sentiment	0.0005	0.01	0.03	0.14	0.14	0.99
Fear	0.0003	0.002	0.004	0.007	0.007	0.41
Joy	0.0006	0.005	0.01	0.07	0.03	0.99
Anger	0.0004	0.0012	0.003	0.01	0.01	0.85

**Table 2 vaccines-10-01713-t002:** All data and their sources used in the study.

Data	Source
Spanish Facebook Data	CrowdTangle [26]
Health Literacy (HL)	University of North Carolina at Chapel Hill [42]
Social Vulnerability Index (SVI)	Centers for Disease Control and Prevention CDC/ATSDR SVI Database [43]
Vaccination Rate	Texas COVID-19 Hospital Resource Usage and Vaccinations [27]
Race and Ethnicity Composition	American Community Survey (ACS) demographic and housing estimates of the US Census Bureau [29]
Sentiments/Emotions	Pysentimiento models, fine-tuned on TASS 2020 [30,31,37] and EmoEvent [38]

**Table 3 vaccines-10-01713-t003:** The association between vaccination increase and negative sentiment.

Variable	Estimate	Standard Error	*p*-Value
Negative sentiment	−5552.98	2228.39	0.013
HL	768.19	3578.13	0.83
SVI	−1902.99	90,091.50	0.98
% Hispanic	103,565.74	216,499.41	0.63
% Black	622,821.87	328,903.21	0.06
% Asian	−61,849.82	563,043.91	0.91

**Table 4 vaccines-10-01713-t004:** The association between vaccination increase and fear.

Variable	Estimate	Standard Error	*p*-Value
Fear	−112,303.83	33,655.89	0.0008
HL	758.37	3562.77	0.83
SVI	−1066.09	90,069.23	0.99
% Hispanic	102,014.19	215,582.03	0.64
% Black	619,033.07	327,611.85	0.06
% Asian	−57,806.59	563,194.91	0.92

## Data Availability

Not applicable.

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
