# Peer review of "Spanish Facebook Posts as an Indicator of COVID-19 Vaccine Hesitancy in Texas"

_vaccines, 2022, doi:10.3390/vaccines10101713_

Round 1

Reviewer 1 Report

This is a great article; there is not much to revise. 

I would perhaps have some captured Facebook posts as examples in the Appendix section and also could elaborate a little more regarding data cleaning for the sentiment analysis.

Author Response

Comments and Suggestions for Authors
This is a great article; there is not much to revise.
I would perhaps have some captured Facebook posts as examples in the Appendix section and also could elaborate a little more regarding data cleaning for the sentiment analysis.

Thank you for the suggestion to add some example posts from the dataset. We included five examples in Appendix B, at the end of the manuscript (lines 598-621). We appreciate the reminder to explain data cleaning in more detail. Hence, we added a paragraph about text pre-processing (data cleaning) in Section 2.2. Dependent, Independent, and Control Variables, line numbers 283-296.

Reviewer 2 Report

I have revised the article sent for review. The use of social network analysis to know the state of the art of a certain subject such as COVID vaccination in this case is highly interesting. 

I think the authors should discuss more in the text some aspects that for those of us who are not knowledgeable about the subject may be difficult to understand: how the Spanish texts on Facebook can conclude that they are Hispanic inhabitants of Texas and not from another geographic region.

Another topic, which I think it would be worthwhile to dedicate more space to how a current of opinion can be displaced or modified through the use of Facebook manipulation.

I believe that with these modifications, the text could be evaluated for publication. 

Author Response

Comments and Suggestions for Authors
I have revised the article sent for review. The use of social network analysis to know the state of the art of a certain subject such as COVID vaccination in this case is highly interesting.
I think the authors should discuss more in the text some aspects that for those of us who are not knowledgeable about the subject may be difficult to understand: how the Spanish texts on Facebook can conclude that they are Hispanic inhabitants of Texas and not from another geographic region.

authors reply: We are aware that some experts are not very familiar with how social media data are structured and what kind of information data provide. Therefore, we appreciate your suggestion to clarify this important aspect of the paper to the readers. Thus, we added a paragraph about the data collection of posts related to geographic regions in section 2.1.1 Facebook Data Collection, page 4, lines 172-178.

Another topic, to which I think it would be worthwhile to dedicate more space to how a current of opinion can be displaced or modified through the use of Facebook manipulation.

authors reply: As this is one of the main points made in the study, we agree that it is relevant to dedicate more space to it. We added a few sentences in the 4. Discussion section (page 11, lines 429-434) explaining in more detail the use of social media to reduce vaccine hesitancy.

I believe that with these modifications, the text could be evaluated for publication.

Reviewer 3 Report

global pandemic’ - remove ‘global’, as ‘pan’ in ‘pandemic’ means ‘global’. ‘For example, a recent study found that’ – remove, check throughout the manuscript for unnecessary content. Research questions and hypotheses must be constructed based on more specific supporting sources, preferably as recent as possible. ‘on testing the following hypothesis’ – you mean ‘hypotheses’, as there are 5. Sometimes, CrowdTangle is misspelled (CrowdTanle). Try and use the proper word for ‘it’. For example, ‘Even though the data contains certain limitations, it still’ – data must be used in plural. ‘Furthermore, at the end of July, it was shown that individuals who were infected with COVID-19 in the past were more likely to experience adverse events following immunization with the Pfizer vaccine [27].’ - Try and provide more references to support your ideas that are typically substantiated by only one source – and as recent as possible. You should compare your results with others in terms of concrete data for better research integrative value. ‘The main strength and novelty of this study is’ – are, not is. ‘the utilization of social media and the focus on Spanish language posts, which provides’ – provide. Check the entire manuscript for such errors. The conclusion, too short, should clarify the main contribution of the paper and the value added to the field. Please provide more details regarding the study limitations and strengths and what this means for the study findings. The manuscript has a very low integrative value in the current debates on the topic: the reference list includes only six references from peer reviewed journals.
The relationship between pervasive misinformation and preventive behavior adherence as regards COVID-19 vaccine hesitancy has not been covered, and thus such recent sources can be cited:
Morris, K. (2021). “COVID-19 Vaccine Hesitancy: Misperception, Distress, and Skepticism,” Review of Contemporary Philosophy 20: 105–116. doi: 10.22381/RCP2020216.
Pera, A., and Cuțitoi, A.-C. (2021). “COVID-19 Vaccine Education: Vaccine Hesitancy Attitudes and Preventive Behavior Adherence,” Analysis and Metaphysics 20: 62–73. doi: 10.22381/am2020214.
Morgan, V., Zauskova, A., and Janoskova, K. (2021). “Pervasive Misinformation, COVID-19 Vaccine Hesitancy, and Lack of Trust in Science,” Review of Contemporary Philosophy 20: 128–138. doi: 10.22381/RCP2020218.

Author Response

Thank you for reviewing our manuscript!

We fixed grammatical errors everywhere in the paper and expanded the conclusion by adding suggested points. We also added 13 different references to the reference list and a few sentences in the introduction citing these references.

Round 2

Reviewer 3 Report

Unfortunately, the authors have not replied point-by-point to the specific comments I made, but provided some general response. This is very unusual and unacceptable.

Author Response

Reviewer 3

Comments and Suggestions for Authors

global pandemic’ - remove ‘global’, as ‘pan’ in ‘pandemic’ means ‘global’. ‘For example, a recent study found that’ – remove, check throughout the manuscript for unnecessary content. Research questions and hypotheses must be constructed based on more specific supporting sources, preferably as recent as possible. ‘on testing the following hypothesis’ – you mean ‘hypotheses’, as there are 5. Sometimes, CrowdTangle is misspelled (CrowdTanle). Try and use the proper word for ‘it’. For example, ‘Even though the data contains certain limitations, it still’ – data must be used in plural. ‘Furthermore, at the end of July, it was shown that individuals who were infected with COVID-19 in the past were more likely to experience adverse events following immunization with the Pfizer vaccine [27].’ - Try and provide more references to support your ideas that are typically substantiated by only one source – and as recent as possible. You should compare your results with others in terms of concrete data for better research integrative value. ‘The main strength and novelty of this study is’ – are, not is. ‘the utilization of social media and the focus on Spanish language posts, which provides’ – provide. Check the entire manuscript for such errors.

Thank you for catching so many grammatical and spelling errors. We fixed such errors everywhere in the manuscript.

 The conclusion, too short, should clarify the main contribution of the paper and the value added to the field. Please provide more details regarding the study limitations and strengths and what this means for the study findings.

 We appreciate this insight, as the conclusion should highlight the study findings, contributions, and limitations. Therefore, we expanded the conclusion by adding suggested points, including the main study findings, value added to the field, strengths, and limitations of the study (edits made on page 13 in the Conclusion section).

The manuscript has a very low integrative value in the current debates on the topic: the reference list includes only six references from peer-reviewed journals. The relationship between pervasive misinformation and preventive behavior adherence as regards COVID-19 vaccine hesitancy has not been covered, and thus such recent sources can be cited:

Morris, K. (2021). “COVID-19 Vaccine Hesitancy: Misperception, Distress, and Skepticism,” Review of Contemporary Philosophy 20: 105–116. doi: 10.22381/RCP2020216.

Pera, A., and Cuțitoi, A.-C. (2021). “COVID-19 Vaccine Education: Vaccine Hesitancy Attitudes and Preventive Behavior Adherence,” Analysis and Metaphysics 20: 62–73. doi: 10.22381/am2020214.

Morgan, V., Zauskova, A., and Janoskova, K. (2021). “Pervasive Misinformation, COVID-19 Vaccine Hesitancy, and Lack of Trust in Science,” Review of Contemporary Philosophy 20: 128–138. doi: 10.22381/RCP2020

Thank you for reviewing our manuscript and providing detailed feedback for improvement. We agree that a research manuscript should include more relevant peer-reviewed references. See below the references added to address this issue. We also added in the Discussion Section the following statement:

In addition to social determinants of health and health literacy, particular psychological determinants, such as beliefs, and attitudes, should be integrated into future studies, as the relationship between misinformation and preventive behavior adherence also plays a role in COVID-19 vaccine hesitancy [added the 3 references suggested by the reviewer as 46-48] (lines 419-423).

Additional references added:

  • Added a reference in the Introduction – lines 35 and 36, page 1.
  • Added 6 references explaining the reasons behind vaccine hesitancy in the Introduction. Page 2, lines 50-52.
  • Added 5 references to the studies that explored the association between sentiments of Twitter posts in the context of the pandemic (Introduction section, line 54).
  • Added the proposed references to the Discussion section, lines 423.

Thank you for helping us improve the quality of our manuscript, and I hope these revisions satisfy your concerns.

Round 3

Reviewer 3 Report

The revised version can be published.